# The Effectiveness of a Combined Healthy Eating, Physical Activity, and Sleep Hygiene Lifestyle Intervention on Health and Fitness of Overweight Airline Pilots: A Controlled Trial

**DOI:** 10.3390/nu14091988

**Published:** 2022-05-09

**Authors:** Daniel Wilson, Matthew Driller, Paul Winwood, Tracey Clissold, Ben Johnston, Nicholas Gill

**Affiliations:** 1Te Huataki Waiora School of Health, The University of Waikato, Hamilton 3216, New Zealand; nicholas.gill@waikato.ac.nz; 2Faculty of Health, Education and Environment, Toi Ohomai Institute of Technology, Tauranga 3112, New Zealand; paul.winwood@toiohomai.ac.nz (P.W.); tracey.clissold@toiohomai.ac.nz (T.C.); 3Sport and Exercise Science, School of Allied Health, Human Services and Sport, La Trobe University, Melbourne 3086, Australia; m.driller@latrobe.edu.au; 4Sports Performance Research Institute New Zealand, Auckland University of Technology, Auckland 1010, New Zealand; 5Aviation and Occupational Health Unit, Air New Zealand, Auckland 1142, New Zealand; ben.johnston@otago.ac.nz; 6New Zealand Rugby, Wellington 6011, New Zealand

**Keywords:** weight loss, nutrition, fruit and vegetable intake, aerobic capacity, moderate-to-vigorous physical activity, lifestyle medicine

## Abstract

(1) Background: The aim of this study was to evaluate the effectiveness of a three-component nutrition, sleep, and physical activity (PA) program on cardiorespiratory fitness, body composition, and health behaviors in overweight airline pilots. (2) Methods: A parallel group study was conducted amongst 125 airline pilots. The intervention group participated in a 16-week personalized healthy eating, sleep hygiene, and PA program. Outcome measures of objective health (maximal oxygen consumption (VO_2max_), body mass, skinfolds, girths, blood pressure, resting heart rate, push-ups, plank hold) and self-reported health (weekly PA, sleep quality and duration, fruit and vegetable intake, and self-rated health) were collected at baseline and post-intervention. The wait-list control completed the same assessments. (3) Results: Significant group main effects in favor of the intervention group were found for all outcome measures (*p* < 0.001) except for weekly walking (*p* = 0.163). All objective health measures significantly improved in the intervention group when compared to the control group (*p* < 0.001, *d* = 0.41–1.04). Self-report measures (moderate-to-vigorous PA, sleep quality and duration, fruit and vegetable intake, and self-rated health) significantly increased in the intervention group when compared to the control group (*p* < 0.001, *d* = 1.00–2.69). (4) Conclusion: Our findings demonstrate that a personalized 16-week healthy eating, PA, and sleep hygiene intervention can elicit significant short-term improvements in physical and mental health outcomes among overweight airline pilots. Further research is required to examine whether the observed effects are maintained longitudinally.

## 1. Introduction

Adverse health outcomes promoted by occupational demands of airline pilots including shift and irregular work schedules, circadian disruption, sedentary activity, and high fatigue [1] may be mitigated through attainment of health guidelines for lifestyle behaviors: healthy diet, physical activity (PA), and sleep [2,3]. Non-communicable diseases (NCDs) including cardiovascular disease (CVD), stroke, type 2 diabetes, and their major risk factors are among leading causes of mortality and morbidity worldwide [4]. The presence of modifiable behavioral NCD risk factors including obesity, hypertension, physical inactivity, low cardiorespiratory fitness, unhealthy dietary patterns, short sleep, depression, high perceived stress levels, and high fatigue are each associated with adverse outcomes to acute and chronic health [4,5,6]. Obesity is a complex, widespread, yet modifiable NCD risk factor that poses a significant public health threat [7]. The obesity prevalence worldwide was estimated as 13% in 2015, which is nearly double the prevalence from 1980 [7]. In 2020, 67% of male airline pilots in New Zealand were classified as overweight or obese with hypertension affecting 27% of the population [8]. Moreover, this study reported the prevalence of insufficient fruit and vegetable intake, physical inactivity, and <7 h sleep per night among airline pilots as 68%, 48%, 33.5%, respectively [8].

The global economic burden associated with NCDs is estimated as $47 trillion between 2010 and 2030 [4]. Previous research has demonstrated evidence of significantly reduced longitudinal health care cost utilization following diet and exercise lifestyle interventions [9]. Relevantly, airline pilots undergo annual or biannual medical examinations, results of which influence flight certification status [10]. Ongoing health care costs associated with the presence of NCDs and their risk factors present economic implications for aviation medical care [4,10].

Better health status is generally associated with enhanced productivity and work performance [11]. In the context of commercial aviation, pilot work performance is imperative to flight operation safety. As established in the International Civil Aviation Organization’s Annex 1, aviation medicine providers are required to implement appropriate health promotion for license holders (pilots) to reduce future medical risks to flight safety [10]. Thus, interventions that promote positive health of pilots, mitigate health risk factors for NCDs, and reduce longitudinal health care costs of employees are of importance to aviation medicine, health practices, and policies.

Limited studies have investigated the efficacy of health promotion interventions among airline pilots, and no studies to date have reported on cardiorespiratory fitness or body fat percentage among this occupational group [1]. Based on the findings of our recent preliminary research [2,12], we found a personalized three-component healthy eating, sleep hygiene, and PA intervention produced favorable outcomes in subjective health and reductions in body mass and blood pressure among airline pilots. Utilizing a different sample of pilots, the aim of the present study was to evaluate the effects of a three-component healthy eating, sleep hygiene, and PA program on cardiorespiratory fitness, body composition, and health behaviors in overweight airline pilots. It was hypothesized that the intervention group would have significantly greater improvements in physical fitness, body composition and health behaviors compared to the wait-list control group at four months.

## 2. Materials and Methods

### 2.1. Design

A parallel controlled study (intervention and control) with pre- and post-testing was conducted to evaluate the effectiveness of a personalized three-component, 16-week lifestyle intervention for enhancing subjective and objective health indices in airline pilots. This study was approved by the Human Research Ethics Committee of the University of Waikato in New Zealand; reference number 2020#07. The trial protocol is registered at The Australian New Zealand Clinical Trials Registry (ACTRN12622000233729).

### 2.2. Participants

The participants comprised of self-selected airline pilots who were recruited from a large international airline in New Zealand. Invitations to participate in the study were distributed to all airline pilots within the company through internal communication networks. Group allocation was determined by a first in, first serve basis due to intervention implementation capacity. Accordingly, pilots who expressed interest to participate in the study early and satisfied the eligibility criteria were allocated to the intervention group (*n* = 86) and subsequent enrolments that exceeded initial capacity were allocated to the wait-list control (*n* = 80). Participants involved pilots from short-haul (regional flights) and long-haul (international flights) rosters. The participants allocated to the wait-list control group received no intervention and were invited to participate in the intervention after the study period.

Potentially eligible pilots who volunteered to participate were screened according to the following eligibility criteria: (a) aged >18 years, (b) pilots with a valid commercial flying license, (c) working on a full-time basis, (d) having a body mass index (BMI) of ≥25 (overweight), and (e) a resting blood pressure of >120/80 (systolic/diastolic). Pilots were excluded if medical clearance was deemed necessary prior to engagement in a PA program after completion of the 2020 Physical Activity Readiness Questionnaire for Everyone (PAR-Q+) [13].

Informed consent was obtained from participants prior to commencement of participation in the study and participants were notified that they were permitted to withdraw at any time during the study if they wish to do so. To encourage data blinding and anonymity during data analysis, participants were allocated a unique identifier code on their informed consent form and were instructed to input this into their online health survey in lieu of their name.

### 2.3. Intervention

At baseline the intervention group completed an individual face-to-face 60-min consultation session with an experienced health coach practitioner located at the airline occupational health facility, followed by provision of a personalized health program. Participants also received weekly educational content emails throughout the intervention and a mid-intervention follow-up phone call with a health coach to discuss progress and support adherence. Health coaching advice delivered to pilots was evidence-based and derived from experts in the fields of dietetics, physical activity, and sleep science.

For extended details of the procedures associated with the three-component intervention, readers are referred to the study of Wilson and colleagues [12]. In brief, the intervention incorporated seven behavior change techniques (BCT) including collaborative goal setting, action planning, problem solving, information about health consequences, self-monitoring, feedback on behaviors, and reviewing of outcomes. The intervention utilized 35 participant interactions: including two face-to-face consultations (baseline and post-intervention), one mid-intervention telephone call, 16 weekly emails, and 16 weekly self-monitoring surveys.

Between the participant and health coach, personalized collaborative outcome, process, and performance goals [12] were established at baseline for (a) sleep hygiene, (b) healthy eating, and (c) PA. Healthy eating goals were defined based on a healthy eating resource (see Appendix A, adapted from Beeken and colleagues [14] with amendments derived from Cena and Calder [15]). Sleep goals were set based on a Sleep Hygiene Checklist (see Appendix B) which was derived from previous sleep hygiene and stimulus control studies [12]. Physical activity prescription goals were established based on assessment of individual barriers and facilitators to physical activity, implementation of the frequency, intensity, time, and type principles [16], and progression to fulfillment of sufficient moderate-to-vigorous-intensity physical activity (MVPA) to meet World Health Organization health guidelines [17] according to individual capabilities. Sufficient physical activity was defined as ≥150 min moderate-intensity, or ≥75 min vigorous-intensity, or an equivalent combination MVPA per week [17].

### 2.4. Outcome Measures

Objective measures of health (maximal oxygen consumption (VO_2max_), body mass, skinfolds, girths, blood pressure, resting heart rate, pushups, plank hold) and self-report measures (weekly PA, sleep quality and duration, fruit and vegetable intake, and self-rated health) were collected at baseline and 4 months (post-intervention). Self-report measures (weekly MVPA, sleep duration, fruit, and vegetable intake) were also collected weekly to monitor intervention adherence via an online survey delivered through Qualtrics software (Qualtrics, Provo, UT, USA).

Participants were instructed to avoid large quantities of food, stimulants such as caffeine, and strenuous exercise 4 h prior to measurement of physiological outcome measures. Outcome measurement protocols for body mass, blood pressure, and subjective health have been previously described in detail [2,12]. In brief, at the start of the consultation session, participants completed an electronic questionnaire via an iPad (Apple, California, CA, USA) to provide data for self-report measures. Using standardized methods previously described [2], resting heart rate was measured utilizing a Rossmax pulse oximeter SB220 (Rossmax Taipei, Taiwan, China), height was recorded with SECA 206 height measures, body mass was measured with SECA 813 electronic scales (SECA, Hamburg, Germany), and blood pressure was measured with an OMRON HEM-757 device (Omron Corporation, Kyoto, Japan).

Skinfold measurements were collected following standardized procedures of the International Society for the Advancement of Kinanthropometry (ISAK) [18]. The skinfold sum was determined by measurements obtained for eight locations: biceps, triceps, subscapular, abdominal, supraspinale, iliac crest, mid-thigh, and medial calf. All skinfold measurements were taken from the right side of the body twice, with a third measurement taken if the difference between recordings were greater than 4%. The anthropometrical technical errors were under the recommended limits [18] for all final recorded measurements. Skinfold measurements were conducted by an accredited ISAK anthropometrist, using Harpenden calipers (British Indicators, Hertfordshire, UK) which were sufficiently calibrated as per the manufacturers’ guidelines. Body fat percentage was derived from skinfold assessments and was calculated using updated sex and ethnicity specific equations reported elsewhere [19]. Girth measurements for the waist and hip locations were measured with a thin-line metric tape measure (Lufkin; Apex Tool Group, Sparks, MD, USA) congruent with standardized technique [20].

Push-ups and the plank isometric hold were utilized as assessments of musculoskeletal fitness, using previously reported standardized methods [21,22]. For push-ups, the hand release technique was utilized, where participants were instructed to keep their torso tight so that the shoulders, hips, knees, and ankles were aligned throughout the range of motion. At the bottom position, the hands were lifted from the floor between each push-up. Push-up cadence was coordinated by a metronome and participants completed maximum full range of motion repetitions until the onset of failure to maintain correct form [21]. The basic plank isometric hold technique was utilized, consisting of the participant holding a prone bridge position supported by their feet and forearms. Elbows were below the shoulders with the forearms and fingers extending forward. The neck was maintained in a neutral position so that the body remained straight from the head to the heels. Time was recorded from initiation of the position until the loss of the plank position [22].

For quantification of aerobic fitness, estimated VO_2max_ was obtained by participants performing a previously validated [23,24] 3-min aerobic test (3mAT) on a Wattbike (Woodway USA, Waukesha, WI, USA) electro-magnetically and air-braked cycle ergometer. Participants were given a full explanation of the protocol, safety procedures, the Wattbike seat and handle were fitted appropriately for the participant, who was also fitted with a Polar H10 heart rate strap (Polar Electro, Kempele, Finland). Full details on the procedure have been detailed elsewhere [23]. Participants completed a 10-min warmup consisting of self-paced cycling at 70–90 rpm with two 6-s sprints within that timeframe, as suggested by the manufacturer. The goal of the 3mAT was to maintain the highest power output possible for 3 full minutes. Verbal encouragement was provided, and participants were allowed to adjust the resistance and pedal cadence as needed throughout the test. Each participant’s customized setup was noted, and the same procedures were carried out for the retest at 4 months.

Prior to baseline testing, the Wattbike was calibrated by the manufacturer, and a between session reliability assessment was conducted with the Wattbike utilizing a convenience sample of seven untrained airline pilots (aged = 42 ± 12 years, body mass = 80 ± 11 kg, height = 173 ± 4 cm, mean ± standard deviation (SD), 5 males, 2 females). Following standardized procedures [23,24], participants of the reliability trial performed the 3mAT twice separated by >48 h between assessments. For measurement of estimated VO_2max_, the reliability trial produced a coefficient of variation (CV) of 4.3% and an intraclass correlation coefficient (ICC) of 0.98 (0.90–0.99), denoting acceptable CV [25] and excellent ICC reliability [26].

Self-report measures (PA, sleep quality and duration, fruit and vegetable intake, and self-rated health) have been previously described in detail [12]. In brief, self-rated health (physical and mental) were measured utilizing the Short Health Form 12v2 (SF-12v2) [27]. The International Physical Activity Questionnaire Short Form (IPAQ) was utilized to quantify self-report MVPA [28]. Self-report subjective sleep quality and duration were measured with the Pittsburgh Sleep Quality Index (PSQI) [29]. Daily fruit and vegetable intake were measured using dietary recall questions derived from the New Zealand Health Survey [28].

### 2.5. Statistical Analyses

G-Power software was utilized to calculate sample size required to detect a clinically significant change in primary outcome measures of ≥5% weight loss and a change of 3.5 mL/kg/min for VO_2max_ [30]. Our sample size power calculation suggested 65 pilots were required in each group to achieve 90% power and a 5% significance criterion to detect relevant differences between the intervention and wait-list control groups. To account for 20% dropout observed in a similar study [2], our target sample size was 156.

Statistical Package for the Social Sciences (SPSS, version 28; IBM Corp., Armonk, NY, USA) was utilized for all analyses. Listwise deletion (i.e., entire case record removal) was applied if individual datasets had missing values or for participants who did not complete post-tests. Stem and leaf plots were inspected to ascertain whether there were any outliers in the data for each variable. A Shapiro–Wilk test (*p* > 0.05) and its histograms, Q–Q plots, and box plots were analyzed for the normality of data distribution for all variables. Levene’s test was used to test homogeneity of variance.

Independent *t*-tests were utilized to calculate whether any significant differences existed between groups at baseline. For categorical variables (long haul and short haul) the Chi square test was used. Between group analysis of pre-test and post-test were assessed using paired *t*-tests and analysis of covariance (ANCOVA) (respectively). To control for baseline differences between groups, baseline data were included as a covariate in the ANCOVA [31], in addition to inclusion of age and sex. Effect sizes were calculated using Cohen’s d to quantify between-group effects from pre-test to post-test. Effect size thresholds were set at >1.2, >0.6, >0.2, and <0.2, which were classified as large, moderate, small, and trivial, respectively [32]. The α level was set at a p value of less than 0.05.

## 3. Results

### 3.1. Characteristics of the Study Population

Two-hundred twelve airline pilots were considered for eligibility and 148 were recruited to participate (Figure 1). Of them, 84% (*n* = 125) of recruits provided data for both timepoints, which comprised a combination of short-haul and long-haul rosters (*n* = 60 and 65, respectively). The dropout rates from baseline to post-intervention were 12% (time commitment *n* = 5; ceased employment *n* = 3; testing not fully completed *n* = 1) and 19% (time commitment *n* = 8; ceased employment *n* = 4) for the intervention and wait-list control groups, respectively. As displayed in Table 1, at baseline both groups demonstrated similar characteristics for most health parameters, yet the wait-list control group had lower SBP (t(123) = 1.191, *p* = 0.03, *d* = 0.39) and lower MAP (t(123) = 2.113, *p* = 0.03, *d* = 0.38). No significant differences were observed between groups for sex and fleet type.

### 3.2. Intervention Adherence

For the intervention group, compliance was measured mid-intervention for health behaviors, including self-report weekly MVPA, daily fruit and vegetable intake and average sleep duration per night. Sixty-four (97%) were achieving ≥5 serves of fruit and vegetables per day, 94% reported sleeping ≥7 h sleep per night, and 97% were obtaining ≥150 MVPA (min) per week. Comparatively, 36% of the wait-list control group were achieving ≥5 serves of fruit and vegetables per day, 71% were sleeping ≥7 h per night, and 53% were obtaining ≥150 MVPA (min) per week.

### 3.3. Body Mass, Skinfolds, Waist Girth, Bodyfat Percentage, Blood Pressure and Pulse

Significant group main effects (*p* < 0.001) in favor of the intervention group were found for all variables. Small to large effect size differences were observed from baseline to post-intervention (Table 2). The within-group analysis revealed that the intervention elicited significant improvements (*p* < 0.001) in all measures at post-intervention associated with moderate to large effect sizes (Table 2; Figure 2). The wait-list control group reported a significantly lower body mass (t(57) = 2.538, *p* = 0.014, d = 0.33) and reduced waist girth (t(57) = 2.358, *p* = 0.022, *d* = 0.31), yet no significant changes were observed in other measures.

### 3.4. VO_2max_, Pushups and Plank Hold

Significant group main effects were found for all measures (*p* < 0.001) in favor of the intervention group. The within-group analysis reported significantly greater improved changes from baseline to post-intervention for all physical performance measures in the intervention group (*p* < 0.001), associated with large effect sizes (Table 2; Figure 2). In contrast, the wait-list control group significantly increased push-ups (t(57) = 5.323, *p* < 0.001, *d* = 0.69) and plank hold (t(57) = 3.365, *p* = 0.001, *d* = 0.44), yet no significant change was observed for VO_2max_.

### 3.5. Health Behaviors and Self-Rated Health

Significant group main effects in favor of the intervention group were found for all self-report health measures (*p* < 0.001) except for weekly walking minutes (*p* = 0.163). The within-group analysis reported significantly greater improved health changes from baseline to post-intervention for all self-report health measures in the intervention group (*p* < 0.001), associated with moderate to large effect sizes (see Table 2; Figure 2). Further, the wait-list control group significantly improved weekly walking, weekly MVPA, global PSQI score, and Short Health Form 12v2 physical component summary scale score (PCS-12, *p* < 0.001), enhanced fruit and vegetable intake (*p* = 0.008), and increased sleep hours (*p* = 0.020). The significant changes observed within the wait-list control group from baseline to post-intervention were associated with trivial to small effect sizes (see Table 2).

## 4. Discussion

To our knowledge, this study is the first clinical trial that has explored the effects of a lifestyle intervention on physical fitness and body composition measures among airline pilots. This study aimed to promote enhancement in cardiorespiratory and musculoskeletal fitness, body composition, and health behaviors through a personalized intervention on healthy eating, sleep hygiene, and PA.

For most outcome measures, in support of our initial hypothesis the controlled trial revealed significantly higher improvements in the intervention group compared to the wait-list control group. Our findings suggest that a face-to-face health assessment alone with no provision of an intervention may promote small short-term effects for improvements in health behaviors and weight management among airline pilots. Furthermore, the provision of a personalized multicomponent lifestyle intervention may facilitate moderate to large short-term effects for promoting healthy changes in physical fitness, body composition, and health behaviors among airline pilots.

These findings are important for health care professionals and researchers to provide insight regarding the efficacy of lifestyle interventions for promoting health, and to inform practices relating to disease prevention, health promotion, and public health policymaking. Furthermore, in relation to the limited literature base pertaining to three-component sleep, nutrition, and PA interventions and the insufficient depth of health behavior intervention research among airline pilots, our findings provide novel contributions to this field.

Excessive adiposity is evidently associated with higher all-cause mortality and elevated risk of cardiometabolic NCDs [33]. Counteractively, clinically significant improvements in NCD risk factors have been reported with as little as 2–3% of weight loss among those with high BMI [34]. A meta-analysis of 59 lifestyle weight loss interventions reported a pooled mean weight loss range of 5–8.5 kg (5–9% body mass) within the initial six months, and among studies exceeding 48 months a mean weight loss range of 3–6 kg (3–6% body mass) [35]. Comparatively, in our intervention group we observed 6% weight loss and 1.6 reduction in BMI at four months. Weight loss and BMI alone as assessments of body composition change are inherently limited due to their inability to precisely measure central adiposity, fat distribution, bone density, and lean mass [36].

In the present study we assessed additional body composition metrics with girth and skinfold measures. Waist circumference has been reported as being strongly associated with all-cause and cardiovascular mortality, with or without adjustment for BMI [36]. Further, skinfold thickness has been reported as a better predictor of body fatness compared to BMI [37]. We found the intervention elicited a decrease of 6 cm waist circumference and 28 mm skinfold thickness sum reduction, which were associated with an overall 3.7% reduction in predicted body fat percentage and a decrease of 8.1 mmHg for systolic blood pressure (SBP). These findings are consistent, yet of higher magnitude than a previous meta-analysis which reported exercise training programs were associated with pooled mean reductions of 5.1 mmHg SBP and 2.2 cm waist girth [38]. This study also reported that reductions in blood pressure (BP) and waist circumference were associated with reduced high-density lipoprotein (HDL) cholesterol and metabolic syndrome risk reduction [38]. Thus, interventions which induce these adaptations are of importance for risk reduction of these well-established NCD risk factors [4].

To our knowledge, our study is the first to report on objective measures of cardiorespiratory capacity among airline pilots. Prospective cohort research suggests exercise capacity is an authoritative predictor of mortality among adults, and an increase of 1 MET (3.5 mL/kg/min) is associated with a 12% CVD risk reduction [30]. A meta-analysis of aerobic exercise training interventions among adults (aged 41 ± 5 y) reported a pooled mean increase in VO_2max_ of 3.5 mL/kg/min (1.9–5.2, 95% confidence interval (CI)), associated with a moderate effect size of 0.6 [39]. In comparison, we observed an increase of 4.5 mL/kg/min within our intervention group, associated with a large effect size which exceeds previously suggested thresholds for clinical relevance [26]. However, future research is required to determine whether these acute adaptations are longitudinally maintained after the brief 16-week intervention.

The intervention promoted significant positive health outcomes for health behaviors and self-rated health, associated with moderate to large effect sizes. Sleep duration increased by 0.6 h in the intervention group, which is a lower magnitude compared with a recent meta-analysis of behavioral interventions to extend sleep length, which reported a pooled increase of 0.8 h per night (0.28–1.31, 95% CI) [40]. In part, this variance may be related to the different nature of interventions, where the present intervention targeted multiple-behavior modification for nutrition, sleep, and PA simultaneously, compared with the individual component focus in other studies (i.e., targeting sleep modification alone) [40].

For weekly MVPA we found the intervention elicited an increase of 72 min/week, which is notably higher than a previous meta-analysis which reported a mean increase of 24 min/week from PA interventions implemented in primary care settings [41]. Similarly, a meta-analysis of behavior interventions to increase fruit and vegetable intake reported a pooled mean increase of 1.1 servings per day [42], which was a lower magnitude of change compared to the increase of 3.6 servings following the present intervention. Notably, a meta-analysis of effective BCTs for promoting PA and healthy eating in overweight and obese adults highlighted the use of goal setting and self-monitoring of behavior as strong predictors of positive short and long-term health behavior change [43]. Congruently, our intervention implemented these components in addition to five other BCTs, which may have contributed to the observed effect sizes of change.

### Strengths and Limitations

A strength of this study is our findings add valuable contribution to a small global literature base pertaining to interventions that include components for each healthy eating, PA, and sleep hygiene. The magnitude of effect sizes for positive health change observed in the intervention may be at least partly attributable to; (a) the implementation of seven BCTs including collaborative goal setting, (b) the personalized multiple-component nutrition, PA and sleep approach, (c) the multimodal intra-intervention communication administered via face-to-face consultations, a telephone call, and regular educational emails, and (d) the potential underlying motivation of airline pilots to improve their health to maintain their aviation medical license.

Potential limitations of this study need to be considered in the interpretation of our findings. Firstly, pilots voluntarily participated in the study via self-selection. Thus, those who enrolled may have exhibited higher readiness and motivation for health behavior change than the general population, which may limit the generalizability of our findings. Secondly, for feasibility of implementation and to minimize participant burden, self-report measures for health behaviors were utilized which inherently possess inferior validity to more invasive objective methods. Accordingly, future research, including measures such as a food frequency questionnaire or photo meal logging for dietary behaviors and actigraphy coupled with heart rate monitoring (e.g., smart watches) for PA and sleep monitoring, would be valuable contributions to increase the validity of findings. Third, although the sex characteristics of our sample are congruent with the general airline pilot population [8], the lack of female participants limits the generalizability of our findings to female populations. Thus, future research should evaluate the effects of the intervention among an ample sample size of females. Finally, the intervention was delivered by an experienced health coach, which presents a barrier to intervention adoption at scale. Future research should evaluate the delivery of interventions using similar procedures via cost-effective and scalable methods, such as online modes of delivery (i.e., smartphone application).

## 5. Conclusions

The personalized 16-week healthy eating, sleep hygiene, and PA intervention implemented in this study elicited significant positive changes associated with moderate to large effects sizes in all main outcome measures at four months follow-up, relative to the wait-list control group. Our findings suggest that the achievement of these three guidelines promotes physical and mental health among overweight airline pilots and these outcomes may be transferrable to other populations. However, there is a need for future research to examine whether the observed effects are longitudinally maintained following the intervention.

## Figures and Tables

**Figure 1 nutrients-14-01988-f001:**
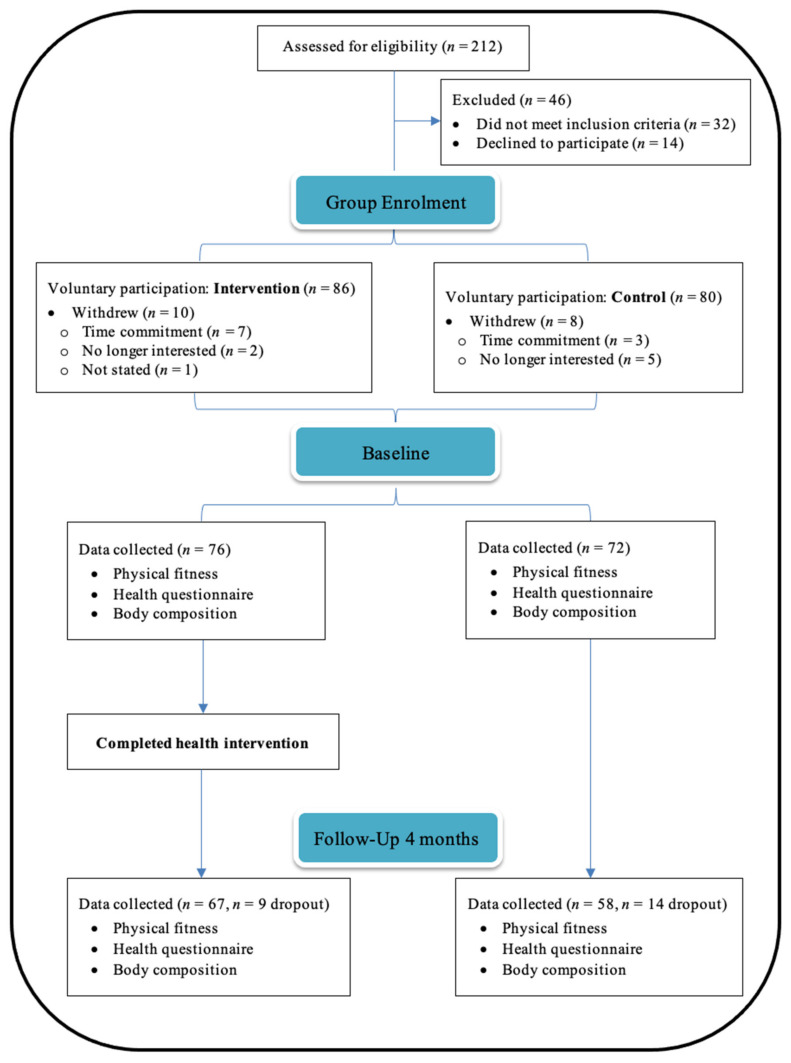
Flow diagram of participant recruitment and data collection.

**Figure 2 nutrients-14-01988-f002:**
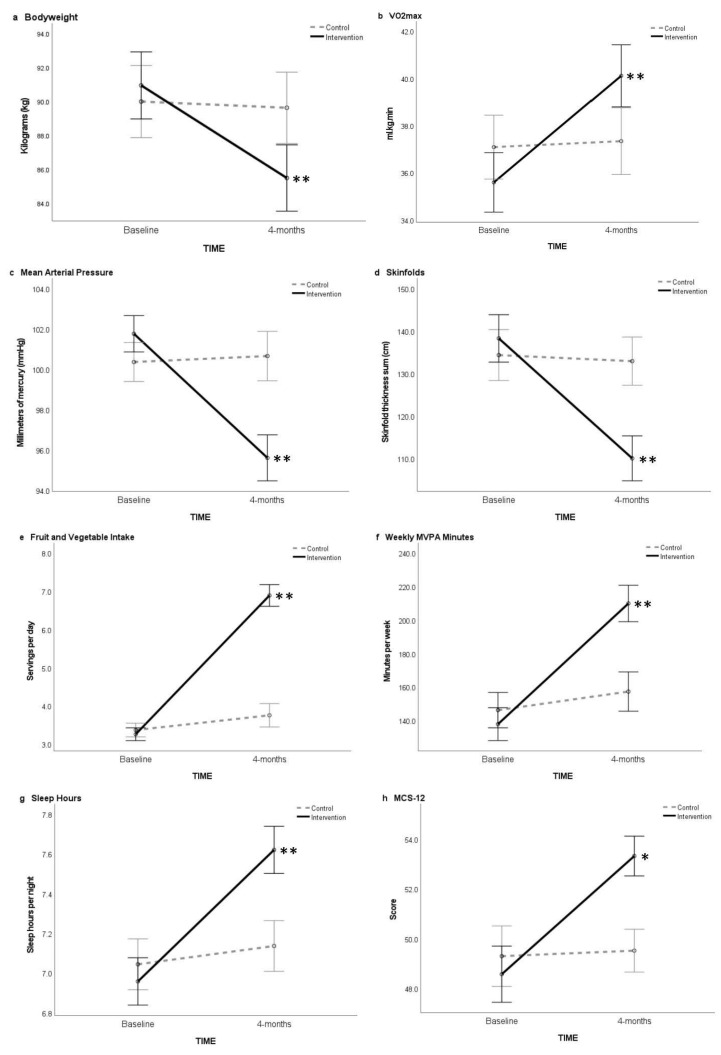
Mean values for health outcomes across time (baseline and 4-months), showing 95% confidence intervals ((**a**), bodyweight; (**b**), VO_2max_; (**c**), Mean Arterial Pressure; (**d**), Skinfolds; (**e**), Fruit and Vegetable Intake; (**f**), Weekly MVPA Minutes; (**g**), Sleep Hours; (**h**), MCS-12). Abbreviations: VO_2max_ = maximal oxygen consumption; MVPA = moderate-to-vigorous physical activity; MCS-12 = Short Health Form 12v2 mental component summary score. Notes: * indicates moderate within group effect size from baseline to 4-months. ** indicates large within group effect size from baseline to 4-months.

**Table 1 nutrients-14-01988-t001:** Baseline characteristics of participants.

Parameters	All subjects (*n* = 125)	Intervention (*n* = 67)	Control (*n* = 58)
Sex (female/male)	12/113	6/61	6/52
Age (years)	44.5 ± 10.7	43.7 ± 10.0	45.6 ± 11.4
Short haul (*n*)	60	34	26
Long haul (*n*)	65	33	32
Height (cm)	178.4 ± 7.4	179.2 ± 6.9	177.4 ± 7.8
Systolic BP (mmHg)	132.3 ± 5.6	133.3 ± 6.0	131.1 ± 4.9 *
Diastolic BP (mmHg)	85.6 ± 3.8	86.0 ± 3.9	85.0 ± 3.6
MAP (mmHg)	101.1 ± 3.8	101.8 ± 3.8	100.4 ± 3.6 *
Pulse (bpm)	66.9 ± 6.6	67.4 ± 6.1	66.4 ± 7.2
Body mass (kg)	90.5 ± 9.2	91.1 ± 8.0	89.8 ± 10.5
BMI (kg/m^2^)	28.4 ± 2.0	28.3 ± 1.7	28.5 ± 3.4
Skinfold sum × 8 sites (mm)	136.5 ± 24.1	138.3 ± 17.7	134.4 ± 29.9
Bodyfat (%)	24.3 ± 3.6	24.7 ± 3.2	23.9 ± 4.0
Waist girth (cm)	96.6 ± 7.6	97.8 ± 8.1	95.2 ± 6.8
Waist to hip ratio	0.93 ± 0.07	0.94 ± 0.07	0.93 ± 0.08
VO_2max_ (mL/kg/min)	36.3 ± 5.4	35.6 ± 5.8	37.0 ± 4.8
Push-ups (repetitions)	17.2 ± 7.3	16.4 ± 6.8	18.1 ± 7.7
Plank hold (s)	79.7 ± 24.7	77.2 ± 25.5	82.5 ± 23.7
Walking per week (min)	73.8 ± 42.5	70.5 ± 32.2	77.7 ± 52.0
MVPA per week (min)	141.8 ± 41.1	138.0 ± 41.6	146.2 ± 40.3
Fruit intake (serve/day)	1.3 ± 0.7	1.5 ± 0.8	1.0 ± 0.6
Vegetable intake (serve/day)	2.0 ± 0.7	1.8 ± 0.7	2.4 ± 0.5
F&V intake (serve/day)	3.3 ± 0.7	3.3 ± 0.7	3.4 ± 0.7
Sleep per day (h)	7.0 ± 0.5	7.0 ± 0.4	7.0 ± 0.6
Global PSQI (score)	6.3 ± 2.1	6.4 ± 2.2	6.1 ± 1.9
MCS-12 (score)	48.9 ± 4.6	48.6 ± 5.8	49.3 ± 2.8
PCS-12 (score)	46.7 ± 3.4	46.3 ± 3.8	47.2 ± 2.8

Note: Mean ± SD reported for all subjects, intervention and control. Abbreviations: SD = Standard deviation; BMI = body mass index; VO_2max_ = maximal oxygen consumption; BP = blood pressure; MAP = mean arterial pressure; MVPA = moderate-to-vigorous physical activity; F&V = fruit and vegetable intake; MCS-12 = Short Health Form 12v2 mental component summary scale; PCS-12 = Short Health Form 12v2 physical health component summary scale; PSQI = Pittsburgh Sleep Quality Index. * Indicates statistical significance (*p* < 0.05).

**Table 2 nutrients-14-01988-t002:** Changes in objective and self-report health measures from baseline to post-intervention at 4-months.

		Intervention			Control			ANCOVA (Group Main Effects)	Between Group ES
		(*n* = 67)			(*n* = 58)				
	Time (Months)	M	SD	Follow Up Change (95% CI)	M	SD	Follow Up Change (95% CI)	*p*	*d*
Body mass (kg)	0	91.1	8.0		89.8	10.5			0.14, Trivial
	4	85.6	7.7	5.5 (4.8–6.1)	89.4	85.6	0.4 (0.1–0.7)	<0.001	−0.41, Small
BMI (kg/m^2^)	0	28.3	1.7		28.5	3.4			0.08, Trivial
	4	26.7	1.6	1.7 (1.5–1.9)	28.4	2.4	0.1 (0.0–0.2)	<0.001	−0.86, Moderate
Systolic BP (mmHg)	0	133.3	6.0		131.1	4.9			0.39, Small
	4	125.2	5.8	8.1 (7.3–8.9)	132.5	5.9	1.3 (0.1–2.8)	<0.001	−1.25, Large
Diastolic BP (mmHg)	0	86.0	3.9		85.0	3.6			0.27, Small
	4	80.8	5.4	5.2 (4.2–6.2)	84.8	4.7	0.2 (0.9–1.4)	<0.001	−0.77, Moderate
MAP (mmHg)	0	101.8	3.8		100.4	3.6			0.38, Small
	4	95.6	5.0	6.2 (5.4–6.9)	100.7	4.7	0.3 (0.8–1.4)	<0.001	−1.04, Moderate
Pulse (bpm)	0	67.4	6.1		66.4	7.2			0.15, Trivial
	4	61.0	6.5	6.3 (4.8–7.8)	67.0	8.8	0.6 (1.0–2.2)	<0.001	−0.78, Moderate
Skinfold sum (mm)	0	138.3	17.7		134.4	29.9			0.16, Trivial
	4	110.1	14.5	28.2 (26–30.5)	133.0	29.8	1.5 (0.5–3.4)	<0.001	−1.00, Moderate
Bodyfat (%)	0	24.7	3.2		23.9	4.0			0.21, Small
	4	21.0	2.8	3.6 (3.3–4.0)	23.7	4.1	0.2 (0.1–0.4)	<0.001	−0.79, Moderate
Waist (cm)	0	97.8	8.1		95.2	6.8			0.35, Small
	4	91.8	7.9	6.0 (5.3–6.8)	94.3	6.9	1.0 (0.1–1.8)	<0.001	−0.34, Small
Waist to hip ratio	0	0.94	0.07		0.93	0.08			0.09, Trivial
	4	0.90	0.07	0.03 (0.02–0.04)	0.92	0.07	0.1 (0.0–0.2)	<0.001	−0.22, Small
VO_2max_ (mL/kg/min)	0	35.6	5.8		37.0	4.8			−0.26, Small
	4	40.2	5.9	4.5 (4.0–5.0)	37.3	5.1	0.2 (0.1–0.6)	<0.001	0.52, Small
Push-ups (repetitions)	0	16.4	6.8		18.1	7.7			−0.22, Small
	4	24.3	7.1	7.8 (6.5–9.1)	19.9	8.1	1.9 (1.2–2.6)	<0.001	0.57, Small
Plank hold (s)	0	77.2	25.5		82.5	23.7			−0.21, Small
	4	120.0	39.6	42.8 (34.4–51.3)	92.1	32.1	9.5 (3.8–15.1)	<0.001	0.77, Moderate
Hours slept (h/day)	0	7.0	0.4		7.0	0.6			−0.17, Trivial
	4	7.6	0.5	0.7 (0.6–0.8)	7.1	0.5	0.1 (0.0–0.2)	<0.001	1.00, Moderate
PSQI Global (score)	0	6.4	2.2		6.1	1.9			0.14, Trivial
	4	4.0	1.3	2.4 (2.0–2.8)	5.8	1.8	0.3 (0.1–0.5)	<0.001	−1.16, Moderate
IPAQ-walk (min)	0	70.5	32.2		77.7	52.0			−0.17, Trivial
	4	97.0	30.0	26.5 (18.1–34.9)	95.4	49.0	17.8 (8.0–27.6)	0.163	0.04, Trivial
IPAQ-MVPA (min)	0	138.0	41.6		146.2	40.3			−0.20, Small
	4	210.3	44.3	72.4 (60.0–84.8)	156.9	46.4	10.8 (5.0–16.5)	<0.001	1.18, Moderate
F&V Intake (serve/day)	0	3.3	0.7		3.4	0.7			−0.17, Trivial
	4	6.9	1.3	3.6 (3.3–4.0)	3.8	0.9	0.4 (0.1–0.7)	<0.001	2.69, Large
PCS-12 (score)	0	46.3	3.8		47.2	2.8			−0.28, Small
	4	51.5	3.4	5.2 (4.4–5.9)	47.9	2.8	0.7 (0.3–1.1)	<0.001	1.14, Moderate
MCS-12 (score)	0	48.6	5.8		49.3	2.8			−0.15, Trivial
	4	53.3	3.6	4.7 (3.7–5.8)	49.5	2.9	0.2 (0.2–0.7)	<0.001	1.15, Moderate

Note: Mean ± SD reported for all participants, intervention and control. Abbreviations: M = mean; SD = standard deviation; CI = Confidence interval; ES = effect size; BMI = body mass index. BP = blood pressure. MAP = mean arterial pressure. MVPA = moderate-to-vigorous physical activity. PSQI = Pittsburgh Sleep Quality Index. IPAQ = International Physical Activity Questionnaire. F&V = fruit and vegetable intake. PCS-12 = Short Health Form 12v2 physical component summary score. MCS-12 = Short Health Form 12v2 mental component summary score.

## Data Availability

Not applicable.

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
