# Peer review of "The Effectiveness of a Combined Healthy Eating, Physical Activity, and Sleep Hygiene Lifestyle Intervention on Health and Fitness of Overweight Airline Pilots: A Controlled Trial"

_nutrients, 2022, doi:10.3390/nu14091988_

Round 1

Reviewer 1 Report

The paper reveals that can elicit significant short-term improvements in physical and mental health outcomes among overweight airline pilots by a personalized healthy eating, physical activity, and sleep hygiene intervention. The analysis process is comprehensive and the article is organized, smooth language and so on. However, there are some issues need to be improved:

  1. Introduction: Supplement the number of global obesity and its severity; Supplementary functional food crop weight loss effect, such as the reference:

https://www.hindawi.com/journals/omcl/2020/3836172/

https://doi.org/10.1080/87559129.2021.2024221

  1. Materials and Methods: If the text can be more refined, it is conducive to reading;
  2. Discussion: If there is difference in obesity between pilots and the general population, please add the reasons for the difference;
  3. Conclusions: It should be appropriately supplemented appropriately.

Author Response

Comments and Suggestions for Authors: The paper reveals that can elicit significant short-term improvements in physical and mental health outcomes among overweight airline pilots by a personalized healthy eating, physical activity, and sleep hygiene intervention. The analysis process is comprehensive and the article is organized, smooth language and so on. However, there are some issues need to be improved:

Author Response: Thank you to reviewer 1 for taking the time to review our manuscript. The edits and suggestions provided below have improved the overall quality of the paper. We have made amendments based on your suggestions, and our individual responses to each of your comments are listed below.

Q1: Introduction: Supplement the number of global obesity and its severity; Supplementary functional food crop weight loss effect, such as the reference:

            https://www.hindawi.com/journals/omcl/2020/3836172/

            https://doi.org/10.1080/87559129.2021.2024221

A1: Thank you for this suggestion, we have made the requested amendment on lines 50-53 as follows: “[4-6]. Obesity is a complex and largely preventable physiological risk factor that negatively affects many systems of the body and comprises a significant public health threat and NCD risk [7]. The global obesity prevalence nearly doubled from 7% in 1980 to 13% in 2015 [7]…”

Q2: Materials and Methods: If the text can be more refined, it is conducive to reading

A2: Thank you for this suggestion, we appreciate it. In construction of this section, we prioritized concise coverage of content and referred to previous works that comprehensively discussed full methodology outlining. Further, we have made some minor amendments to the methods based on feedback from this revision round. If the reviewer has specific recommendations for which content should be refined, we are happy to consider further amendments.

Q3: Discussion: If there is difference in obesity between pilots and the general population, please add the reasons for the difference

A3: Thank you for your comments, we appreciate it. As the present study’s aim was to evaluate the effectiveness of the intervention on health parameters of overweight airline pilots, we do not perceive discussion of contributing factors to the difference in general obesity prevalence between pilots and the general population as pertinent information to the discussion of the present intervention.

However, we are happy to discuss this further and consider adding amendments with further clarification of the relevance to this body of work.

Q4: Conclusions: It should be appropriately supplemented appropriately

A4: Thanks for your comment, if you have specific aspects of the conclusion for which you suggest amendments, we are happy to discuss these.

Reviewer 2 Report

This study presents the results of an intervention of combined healthy eating, physical activity, and sleep hygiene lifestyle on health and fitness outcomes of overweight airline pilots. The study addresses an important problem, that of overweight, sleep and eating schedule disturbances, which have been shown to have negative impacts on health. The study sample is an interesting example in this context, and as stated by the authors, represents a population group which has not well investigated previously for these risks. The intervention study was multi-targeted, and builds upon previous work. The manuscript is clearly written,  well-structured, and follows scientific writing standards. 

Many aspects of the research protocol were based on previous studies, and one of these was published recently in in Nutrients. There are sufficient differences between the studies, such that the submitted manuscript clearly represents an independent study.

Below are the points to address, many of which are quite small.

Abstract – is clear and coherent.  Is the word number within the limit of Nutrients?

Introduction:

A high number of indicators and parameters were included in this study to assess the pilots. Yet the reader is not well informed about their selection. Could the authors explain briefly how these indicators were selected, i.e. main justifications, especially for the ones which might be less familiar to readers of the journal ‘Nutrients’.

Line 59-60 – rephrase the sentence to avoid repeating ‘ongoing’ twice.

Lines 90 – 111 – the characteristics of participants are not fully described, The terms short and long haul appear in Table 1 but no explanation is given about these in the methods.

Line 108 – were made aware that they could withdraw – I would revise this to ‘were informed’ that …

Line 131 – Suggestion: Healthy eating goals were defined based on a resource for healthy eating…

Line 138-1139 – the explanation for MVPA is not very clear. Please re-word.

Line 139  in congruence with individual --- A better wording would be ‘according to’ individual capabilities…

Line 144 – sleep quality and duration -  would seem to be the better term to use, and throughout the manuscript.

Line  160 – skinfold measurements were collected ‘following’ –

Line 186 – until the loss of the plank position.

Line 192 – for the participant, who was also fitted with a Polar H10 heart rate strap.

Line 197-198 – re-arrange the sentence so that ‘throughout the test’ is at the end.

Line 199 – the same procedures were carried out

Line 224 – Could you explain ‘Listwise deletion’ or use another term.

Line 237 – The editor might decide, but here you could use >1.2, >0.6, …

Line 245 – The terms short haul and long-haul rosters were not previously defined or explained.  The reader should not have to check the past paper to find out about these.

Line 152 – fleet type – the same comment as above, please define this term in the methods.

Line 260 – MVPA (min) per week would seem to be more logical.

Line 261 – there is an extra ‘sleep’ in this sentence;

Line 260 -  this sentence which is somewhat awkward, please improve it.  ‘36% of ? reached  ≥5 servings of fruit and vegetables per day.

Line 323-324 – Please re-write and shorten this sentence aiming for clarity.

Line 343 – we assessed additional body composition metrics  - would be better.

Line 364  are there units with the value ‘size of 0.6’ ?

Line 366 – after the brief 16-week intervention – would be better.

Line 374 – this last sentence is a little bit vague, could you be more precise, by listing 2-3 components which were the focus in other studies.

Line 296 – instead of ‘to support’ – I suggest ‘to maintain’.

Line 404  - As an improvement to their method for assessing food quality (frequency of fruits & vegetables), the authors mention weighed food logging’  - indeed this is a very comprehensive dietary assessment method.  However, would this be feasible for this type of intervention study? Weighed food logging is very labour-intensive, Could the authors propose a more realistic indicator of diet, which might be feasible in this type of intervention study, such as a short food frequency questionnaire or another suitable method?

Line 416 – elicited significant positive changes

Figure 2 – could the authors justify the inclusion of this Figure in the manuscript, what additional information is provided to the reader, compared to Table 2?

Author Response

Comments and Suggestions for Authors: This study presents the results of an intervention of combined healthy eating, physical activity, and sleep hygiene lifestyle on health and fitness outcomes of overweight airline pilots. The study addresses an important problem, that of overweight, sleep and eating schedule disturbances, which have been shown to have negative impacts on health. The study sample is an interesting example in this context, and as stated by the authors, represents a population group which has not well investigated previously for these risks. The intervention study was multi-targeted, and builds upon previous work. The manuscript is clearly written,  well-structured, and follows scientific writing standards. 

Many aspects of the research protocol were based on previous studies, and one of these was published recently in in Nutrients. There are sufficient differences between the studies, such that the submitted manuscript clearly represents an independent study. 

Below are the points to address, many of which are quite small.

Author Response: Thank you so much to reviewer 2 for taking the time to review our manuscript. The edits and suggestions provided below have improved the overall quality of the paper. We have made amendments based on your suggestions, and our individual responses to each of your comments are listed below.

Q1: Abstract – is clear and coherent.  Is the word number within the limit of Nutrients?

A1: Thanks for your comment, we have made a few slight amendments while maintaining the integrity of the content to reduce the word count which aligns better now with the word number limit for Nutrients. The updated abstract reads as follows: “(1) Background: The aim of this study was to evaluate the effectiveness of a three-component nutrition, sleep, and physical activity (PA) program on cardiorespiratory fitness, body composition, and health behaviors in overweight airline pilots. (2) Methods: A parallel group study was conducted amongst 125 airline pilots. The intervention group participated in a 16-week personalized healthy eating, sleep hygiene, and PA program. Outcome measures of objective health (VO2max, body mass, skinfolds, girths, blood pressure, resting heart rate, push-ups, plank hold) and self-reported health (weekly PA, sleep quality and quantity, fruit and vegetable intake, and self-rated health) were collected at baseline and post-intervention. The wait-list control completed the same assessments. (3) Results: Significant group main effects in favor of the intervention group were found for all outcome measures (p = < 0.001) except for weekly walking (p = 0.163). All objective health measures significantly improved in the intervention group when compared to the control group (p = < 0.001, d = 0.41–1.04). Self-report measures (moderate-to-vigorous PA, sleep quality and quantity, fruit and vegetable intake, and self-rated health) significantly increased in the intervention group when compared to the control group (p = < 0.001, d = 1.00–2.69). (4) Conclusion: Our findings demonstrate that a personalized 16-week healthy eating, PA, and sleep hygiene intervention can elicit significant short-term improvements in physical and mental health outcomes among overweight airline pilots. Further research is required to examine whether the observed effects are maintained longitudinally.”

Q2: Introduction: A high number of indicators and parameters were included in this study to assess the pilots. Yet the reader is not well informed about their selection. Could the authors explain briefly how these indicators were selected, i.e. main justifications, especially for the ones which might be less familiar to readers of the journal ‘Nutrients’.

A2: Thanks for your advice, we appreciate it. As the parameters of interest are widely reported in the literature and in attempt to keep the manuscript writing concise for readers, can you please advise what specific parameters you require supporting descriptions for and we can integrate these.

Q3: Line 59-60 – rephrase the sentence to avoid repeating ‘ongoing’ twice.

A3: Thanks for pointing this out, this amendment has been completed.

Q4: Lines 90 – 111 – the characteristics of participants are not fully described, The terms short and long haul appear in Table 1 but no explanation is given about these in the methods.

A4: Thanks for your comment, we have added the following description on lines 109-110 to address this: “Participants consisted of pilot rosters including long haul (international flights) and short haul (regional flights).”

Q5: Line 108 – were made aware that they could withdraw – I would revise this to ‘were informed’ that…

A5: Thanks for pointing this out, this amendment has been completed.

Q6: Line 131 – Suggestion: Healthy eating goals were defined based on a resource for healthy eating…

A6: Thanks for pointing this out, this amendment has been completed.

Q7: Line 138-1139 – the explanation for MVPA is not very clear. Please re-word.

A7: Thanks for pointing this out, the following sentence has been added on lines 152-154 to clarify: “Sufficient physical activity was defined as ≥150 minutes moderate-intensity, or ≥75 minutes vigorous-intensity, or an equivalent combination MVPA per week [17].”

Q8: Line 139 in congruence with individual --- A better wording would be ‘according to’ individual capabilities…

A8: Thanks for your advice, this amendment has been completed.

Q9: Line 144 – sleep quality and duration -  would seem to be the better term to use, and throughout the manuscript.

A9: Thanks for your advice, this amendment has been completed throughout the manuscript.

Q10: Line  160 – skinfold measurements were collected ‘following’ –

A10: Thanks for your advice, this amendment has been completed.

Q11: Line 186 – until the loss of the plank position.

A11: Thanks for your advice, this amendment has been completed.

Q12: Line 192 – for the participant, who was also fitted with a Polar H10 heart rate strap.

A12: Thanks for your advice, this amendment has been completed.

Q13: Line 197-198 – re-arrange the sentence so that ‘throughout the test’ is at the end.

A13: Thanks for your advice, the sentences have been amended as follows: “The goal of the 3mAT was to maintain the highest power output possible for 3 full minutes. Verbal encouragement was provided, and participants were allowed to adjust the resistance and pedal cadence as needed throughout the test.”

Q14: Line 199 – the same procedures were carried out

A14: Thanks for your advice, this amendment has been completed.

Q15: Line 224 – Could you explain ‘Listwise deletion’ or use another term.

A15: Thank you for your comment. The sentence has been amended as follows: “Listwise deletion (i.e. entire case record removal) was applied if individual datasets had missing values or for participants who did not complete post-tests.”

Q16: Line 237 – The editor might decide, but here you could use >1.2, >0.6, …

A16: Thanks for your advice, this amendment has been completed.

Q17: Line 245 – The terms short haul and long-haul rosters were not previously defined or explained.  The reader should not have to check the past paper to find out about these.

A17: Thanks for this comment, we have added descriptions for each on lines 111-112 as follows: “Participants consisted of pilot rosters including long haul (international flights) and short haul (regional flights).”

Q18: Line 152 – fleet type – the same comment as above, please define this term in the methods.

A18: This has been amended as per previous response.

Q19: Line 260 – MVPA (min) per week would seem to be more logical.

A19: Thanks for your advice, this amendment has been completed.

Q20: Line 261 – there is an extra ‘sleep’ in this sentence;

A20: Thanks for your advice, this amendment has been completed.

Q21: Line 260 -  this sentence which is somewhat awkward, please improve it.  ‘36% of ? reached  ≥5 servings of fruit and vegetables per day.

A21: Thanks for your comment, the sentence has been amended as follows: “Comparatively, 36% of the wait-list control group were achieving ≥5 serves of fruit and vegetables per day, 71% were sleeping ≥7 hr per night, and 53% were obtaining ≥150 MVPA (min) per week.”

Q22: Line 323-324 – Please re-write and shorten this sentence aiming for clarity.

A22: Thanks for your advice, this amendment has been completed.

Q23: Line 343 – we assessed additional body composition metrics  - would be better.

A23: Thanks for your advice, this amendment has been completed.

Q24: Line 364  are there units with the value ‘size of 0.6’ ?

A24: Thanks for your comment,  yes this sentence was referring to the reported the mean pooled effect size of the value 0.6 which is the Cohen’s d effect size.

Q25: Line 366 – after the brief 16-week intervention – would be better.

A25: Thanks for your advice, this amendment has been completed.

Q26: Line 374 – this last sentence is a little bit vague, could you be more precise, by listing 2-3 components which were the focus in other studies.

A26: Thanks for your advice, the sentence has been modified as follows to address this comment: “In part this variance may be related to the different nature of interventions, where the present intervention targeted multiple-behavior modification for nutrition, sleep, and PA simultaneously, compared with the individual component focus in other studies (i.e. targeting sleep modification alone) [40].”

Q27: Line 296 – instead of ‘to support’ – I suggest ‘to maintain’.

A27: Thanks for your advice, this amendment has been completed.

Q28: Line 404  - As an improvement to their method for assessing food quality (frequency of fruits & vegetables), the authors mention weighed food logging’  - indeed this is a very comprehensive dietary assessment method.  However, would this be feasible for this type of intervention study? Weighed food logging is very labour-intensive, Could the authors propose a more realistic indicator of diet, which might be feasible in this type of intervention study, such as a short food frequency questionnaire or another suitable method?

A28: Thanks for this insight, the sentence has been modified as follows to address this: “Accordingly, future research including measures such as a food frequency questionnaire or photo meal logging for dietary behaviors and actigraphy coupled with heart rate monitoring (e.g. smart-watches) for PA and sleep monitoring would be valuable contributions to increase the validity of findings.”

Q29: Line 416 – elicited significant positive changes

A29: Thanks for your advice, this amendment has been completed.

Q30: Figure 2 – could the authors justify the inclusion of this Figure in the manuscript, what additional information is provided to the reader, compared to Table 2?

A30: We perceive Figure 2 to provide value by visual illustration of trends across key outcome measures and Figure 2 denotes within group effect size changes, which may be useful from a practical viewpoint for practitioners. We are open to discuss the removal of this Figure if the editor deems it necessary to do so.